# Identification of the Ovine Keratin-Associated Protein 2-1 Gene and Its Sequence Variation in Four Chinese Sheep Breeds

**DOI:** 10.3390/genes11060604

**Published:** 2020-05-29

**Authors:** Jianqing Wang, Huitong Zhou, Jon G. H. Hickford, Yuzhu Luo, Hua Gong, Jiang Hu, Xiu Liu, Shaobin Li, Yize Song, Na Ke, Lirong Qiao, Jiqing Wang

**Affiliations:** 1Gansu Key Laboratory of Herbivorous Animal Biotechnology, Faculty of Animal Science and Technology, Gansu Agricultural University, Lanzhou 730070, China; wangjq@st.gsau.edu.cn (J.W.); huitong.zhou@lincoln.ac.nz (H.Z.); Jon.hickford@lincoln.ac.nz (J.G.H.H.); luoyz@gsau.edu.cn (Y.L.); hua.gong@lincoln.ac.nz (H.G.); huj@gsau.edu.cn (J.H.); liuxiu@gsau.edu.cn (X.L.); lisb@gsau.edu.cn (S.L.); songyz@st.gsau.edu.cn (Y.S.); ken@st.gsau.edu.cn (N.K.); qiaolr735757@163.com (L.Q.); 2International Wool Research Institute, Faculty of Animal Science and Technology, Gansu Agricultural University, Lanzhou 730070, China; 3Gene-Marker Laboratory, Faculty of Agriculture and Life Sciences, Lincoln University, Lincoln 7647, New Zealand

**Keywords:** keratin-associated protein 2-1 gene (*KRTAP2-1*), variation, sheep, wool traits

## Abstract

Keratin-associated proteins are important components of wool fibers. The gene encoding the high-sulfur keratin-associated protein 2-1 has been described in humans, but it has not been described in sheep. A basic local alignment search tool nucleotide search of the Ovine Genome Assembly version 4.0 using a human keratin-associated protein 2-1 gene sequence revealed a 399-base pair open reading frame, which was clustered among nine previously identified keratin-associated protein genes on chromosome 11. Polymerase chain reaction–single strand conformation polymorphism analysis revealed four different banding patterns, with these representing four different sequences (*A–D*) in Chinese sheep breeds. These sequences had the highest similarity to human keratin-associated protein 2-1 gene, suggesting that they represent variants of ovine keratin-associated protein 2-1 gene. Nine single nucleotide variations were detected in the gene, including one non-synonymous nucleotide substitution. Differences in variant frequencies between fine-wool sheep breeds and coarse-wool sheep breeds were detected. The gene was found to be expressed in various tissues, with the highest expression level in skin, and moderate expression levels in heart and lung tissue. These results reveal that the ovine keratin-associated protein 2-1 gene is variable and suggest the gene might affect variation in mean fiber diameter.

## 1. Introduction

Keratins and keratin-associated proteins are the main structural components of wool fibers. The keratins form the keratin intermediate filaments, while the keratin-associated proteins form a matrix that cross-links the filaments. The physical and mechanical properties of wool fibers are therefore believed to be in-part determined by the keratins and keratin-associated proteins [1].

The keratin-associated proteins characteristically contain a high proportion of either cysteine, or glycine and tyrosine residues, and they have traditionally been divided into three broad groups: the high-sulfur keratin-associated proteins (containing no more than 30 mole percent of cysteine), the ultra-high-sulfur keratin-associated proteins (containing over 30 mole percent of cysteine), and the high glycine/tyrosine keratin-associated proteins (containing 35–60 mole percent of glycine and tyrosine) [2]. More than 100 keratin-associated protein genes from 28 families have been identified in mammals [3,4,5]. Of these, keratin-associated protein 1, 3, 11, 13, 15, 16, 23, and 27 belong to the high-sulfur keratin-associated protein group [3].

To date, 35 keratin-associated protein genes have been identified in the sheep genome [4,6]. Variation in many ovine keratin-associated protein genes have been reported to be associated with various wool traits, including wool yield [7], fleece weight [8], mean fiber diameter-related traits [9,10], prickle factor [4], mean fiber curvature [11], and mean staple length [9]. This suggests that keratin-associated protein genes may be useful as markers for improving wool traits in sheep, although many of the keratin-associated protein 2 genes remain to be identified in sheep.

Many of the keratin-associated protein families consist of multiple members [2,5]. The keratin-associated protein 2 family possibly has the most family members, with five genes having been reported in humans [5]. The human keratin-associated protein 2 genes are predominantly expressed in the middle and upper keratogenous zone of the hair shaft cortex [12]. To date, a keratin-associated protein 2 gene has not been located in the sheep genome, despite there being a partial deoxyribonucleic acid sequence (GenBank U60024) and two protein sequences BIIIA3A (UniProtKB/Swiss-Prot P02443) and BIIIA3 (UniProtKB/Swiss-Prot P02441) reported several decades ago [13]. The proteins BIIIA3A and BIIIA3 are probably the orthologue of human keratin-associated protein 2-1 and keratin-associated protein 2-3, respectively [5].

The People’s Republic of China is the world’s largest sheep producer and has over 90 recognized sheep breeds. The wool breeds can be broadly classified into three types on the basis of wool characteristics: coarse-wool breeds, mid-micron wool breeds, and fine-wool breeds. The Chinese Merino sheep and Gansu Alpine Fine-wool sheep are representative of the fine-wool sheep breeds, with a characteristic mean fiber diameters of 22 and 21 µm, respectively. Tibetan sheep and Kazakh sheep are two of the three coarse wool sheep breeds, with mean fiber diameters of 43 and 46 µm, respectively [14].

In this animal study, we report the identification of ovine keratin-associated protein 2-1 gene and describe variation in the gene’s nucleotide sequence using polymerase chain reaction–single strand conformation polymorphism analysis. We also compare the expression levels of this gene in a variety of tissues in fine wool sheep and the difference in variant frequencies between the fine-wool sheep and coarse-wool sheep breeds that were investigated.

## 2. Materials and Methods

### 2.1. Ethics Statement

All animal experiments were carried out on the basis of the guidelines for the protection and use of laboratory animals (grant number 2006-398) issued by the Ministry of Science and Technology of the People’s Republic of China, and the experiments were also agreed upon by the Gansu Agricultural University, Lanzhou, the People’s Republic of China.

### 2.2. Study Design and Location

This animal study was carried out using sheep derived from the Yili Prefecture, Tianzhu County, Gannan Prefecture of the People’s Republic of China, with the laboratory work being undertaken in the Gansu Key Laboratory of Herbivorous Animal Biotechnology at Gansu Agricultural University (Anning, Lanzhou, China) or the Beijing Genomics Institute (Beijing, China).

### 2.3. Sheep Investigated and Tissue Samples

A total of 216 sheep randomly selected from two fine-wool sheep breeds and two coarse-wool sheep breeds were investigated. These included two fine-wool breeds: Chinese Merino sheep (*n* = 38) and Gansu Alpine Fine-wool sheep (*n* = 66); and two course-wool breeds: Tibetan sheep (*n* = 62) and Kazakh sheep (*n* = 50). These sheep were located in the Yili Prefecture, Tianzhu County, Gannan Prefecture, and Yili Prefecture, respectively, in the People’s Republic of China. Blood samples from these sheep were collected onto TFN paper (Munktell Filter AB, Falun, Sweden) and then returned to the Gansu Key Laboratory of Herbivorous Animal Biotechnology at Gansu Agricultural University (Lanzhou, China), and genomic deoxyribonucleic acid for polymerase chain reaction amplification was purified using a two-step washing procedure [15].

Three separate two-year-old Gansu Alpine Fine-wool sheep from Tianzhu County of the People’s Republic of China were slaughtered to collect seven tissue samples from each: skin, longissimus dorsi muscle, kidney, lung, spleen, heart, and liver tissue. These samples were immediately frozen in liquid nitrogen and then stored at −80 °C.

### 2.4. Bioinformatics Analysis of the Ovine Genome Sequence

A human keratin-associated protein 2-1 gene sequence (GenBank NM_001123387.1) was used to search the Ovine Genome Assembly Oar_version 4.0 (National Center for Biotechnology Information, Bethesda, MD, USA) using the basic local alignment search tool algorithm [16]. The sequence that shared the greatest similarity with the human keratin-associated protein 2-1 gene sequence was assumed to be the ovine keratin-associated protein 2-1 gene. This putative keratin-associated protein 2-1 gene sequence was matched to the ovine keratin-associated protein 2-1 protein sequence (UniProtKB/Swiss-Prot P02443) using DNAMAN version 5.2.10 (Lynnon, BioSoft, Vaudreuil, Quebec, Canada).

### 2.5. Polymerase Chain Reaction Primers and Amplification Protocol

The polymerase chain reaction amplifications and single-strand conformation polymorphism screening for nucleotide sequence variation described below were undertaken in the Gansu Key Laboratory of Herbivorous Animal Biotechnology, Faculty of Animal Science and Technology, Gansu Agricultural University, People’s Republic of China.

On the basis of the putative ovine keratin-associated protein 2-1 gene sequence identified above, we designed two polymerase chain reaction primers (5′-AACAAGGAATGGCATGAGTC-3′ and 5′-GTTGCTTTATAGGAAAGTGGG-3′) to amplify a 565-bp deoxyribonucleic acid fragment that would include the entire coding region of the putative ovine keratin-associated protein 2-1 gene. These primers were synthesized by the Takara Biotechnology Company Limited (Dalian, China).

Amplification reactions contained the genomic deoxyribonucleic acid on a 1.2 mm punch of TFN paper, 150 µM of each deoxyribonucleoside triphosphate (Takara), 2.5 mM Mg^2+^, 0.5 U of Taq DNA polymerase (Takara), and 0.25 µM of each primer. The thermal profile consisted of an initial denaturation for 2 min at 94 °C, followed by 35 cycles of 30 s at 94 °C, 30 s at 59 °C, and 30 s at 72 °C, with a final extension of 5 min at 72 °C. Amplifications were performed in a 20 µL reaction in S1000 thermal cyclers (Bio-Rad, Hercules, CA, USA).

### 2.6. Screening for Nucleotide Sequence Variation

The potential for sequence variation in the polymerase chain reaction amplicons was investigated by using a single-strand conformation polymorphism analysis. A 0.7 µL aliquot of each amplicon was mixed with 7 µL of loading dye (10 mM ethylenediaminetetraacetic acid, 0.025% xylene cyanol, 0.025% bromophenol blue, and 98% formamide). The samples were denatured at 95 °C for 5 min and then immediately cooled on wet ice before being loaded on 16 × 18 cm, 14% acrylamide/bisacrylamide (37.5:1) (Bio-Rad, Hercules, CA, USA) gels. Electrophoresis was performed using Protean II xi cells (Bio-Rad), at 280 V and 29 °C for 18 h in 0.5× Tris-borate-ethylenediaminetetraacetic acid buffer. Gels were silver-stained using the method described by Byun et al. [17].

### 2.7. Sequencing of Variants and Sequence Analyses

In the study, amplicons from the keratin-associated protein 2-1 gene variants that appeared to have a homozygous form in the single strand conformation polymorphism analysis were directly sequenced in both directions at the Beijing Genomics Institute (Beijing, China). The basic local alignment search tool algorithm [16] was used to search for sequences homologous to those derived from the DNA sequencing in the National Center for Biotechnology Information GenBank databases [18], and DNAMAN (Lynnon, BioSoft, Quebec, Canada) was used for subsequent sequence alignments and translation. MEGA version 7.0 [19] was used to construct a phylogenetic tree.

### 2.8. Reverse Transcription and Quantitative Polymerase Chain Reaction Analysis

The reverse transcription and quantitative polymerase chain reaction analyses were undertaken in the Gansu Key Laboratory of Herbivorous Animal Biotechnology, Faculty of Animal Science and Technology, Gansu Agricultural University, People’s Republic of China.

Total ribonucleic acid from the seven different tissues was extracted using TRIzol reagent (Invitrogen, Carlsbad, CA, USA). The quantity and quality of the extracted ribonucleic acid were assessed by ultraviolet spectrophotometry and 2% agarose gel electrophoresis, respectively.

Reverse transcription was carried out using the Prime Script RT Reagent kit with gDNA Eraser (Perfect Real Time, Takara). The synthesized complementary deoxyribonucleic acid was amplified using a pair of PCR primers (5′-CCAGTGTCCAGCCAGACCAC-3′ and 5′-GGGGACTGCACAGAGACG-3′) located within the keratin-associated protein 2-1 gene coding region. The amplification was carried out under the same conditions and thermal profile was used for the amplification of genomic deoxyribonucleic acid, but the amplified template changed from genomic deoxyribonucleic acid to 0.8 μL of the complementary deoxyribonucleic acid. The beta-actin gene (*ACTB*) was used as an internal reference standard, and amplified using the primers 5′-AGCCTTCCTTCCTGGGCATGGA-3′ and 5′-GGACAGCACCGTGTTGGCGTAGA-3′.

The complementary deoxyribonucleic acid samples were tested for the presence of a transcript for ovine keratin-associated protein 15-1 gene using the polymerase chain reaction primers 5′-ATCTTCCGCAGTCCCTG-3′ and 5′-GATGACCGGCAACTCCT-3′, and the method described in Zhao et al. [20]. They were also tested for potential contamination with genomic deoxyribonucleic acid using a pair of *ADIPOQ* primers (5′-ACAGCGTGGATCTGGGTTC-3′ and 5′-CACAATTCACTTTCGGCTGC-3′), which are located in intron 1 and intron 2, respectively, as described by An et al. [21]. The amplification products were electrophoresed through 1.0% agarose gels and visualized using ethidium bromide staining.

To measure the relative expression levels of keratin-associated protein 2-1 gene in different tissues, reverse transcription quantitative polymerase chain reaction was carried out in triplicate using the 2× ChamQ SYBR qPCR Master system (Vazyme, Nanjing, China) on an Applied Biosystems QuantStudio 6 Flex Real-time PCR System (Thermo Fisher Scientific, Waltham, MA, USA). The thermal profile included an initial denaturation of 30 seconds at 95 °C, followed by 40 cycles of 10 seconds at 95 °C, 30 seconds at 60 °C, and 30 seconds at 72 °C. *ACTB* was utilized as an internal reference standard, and the relative expression levels of the keratin-associated protein 2-1 gene in the seven tissues were calculated using a 2-delta delta C_t_ method. Error bars on the expression level graphs were calculated using a one-way analysis of variance (ANOVA) in IBM SPSS Statistics version 24.0 (IBM China Company Limited, Beijing, China).

## 3. Results

### 3.1. Identification of the Keratin-Associated Protein 2-1 Gene in the Sheep Genome

A basic local alignment search tool algorithm [16] search of the Ovine Genome Assembly version 4.0 using a human keratin-associated protein 2-1 gene coding sequence (GenBank NM_001123387.1) revealed a homologous region (nt 40807705_40808103) on sheep chromosome 11. This region contained a 399 bp open reading frame and had 90% nucleotide identity with the human keratin-associated protein 2-1 gene sequence. In proximity to this region were nine previously reported ovine keratin-associated protein genes [2], the keratin-associated protein genes 3-3, 3-2, 3-1, 1-4, 1-1, 1-2, 1-3, 4-1, and 4-3, in order from the centromere to the telomere (Figure 1).

### 3.2. Identification of the Keratin-Associated Protein 2-1 Gene in the Sheep Genome

Four different polymerase chain reaction–single strand conformation polymorphism analysis patterns (designated *A*, *B*, *C*, and *D*) were found for the ovine keratin-associated protein 2-1 gene (Figure 2). For each sheep investigated, either one pattern or a combination of two patterns was observed. Sequencing of amplicons corresponding to the four unique single strand conformation polymorphism patterns revealed four different deoxyribonucleic acid sequences. All of these sequences were different to, but had over 98% similarity to the keratin-associated protein 2-1 gene sequence identified in the ovine genomic assembly.

Phylogenetic analysis revealed that the predicted amino acid sequences of the four deoxyribonucleic acid sequences had highest similarity to a predicted amino acid sequence from a human keratin-associated protein 2-1 gene sequence (NM_001123387.1) and an ovine keratin-associated protein 2-1 protein sequence (P02443) among all of the high-sulfur keratin-associated protein sequences identified in humans and sheep (Figure 3). This suggests that these four newly identified sequences represent allelic variants of the ovine keratin-associated protein 2-1 gene.

Nine single-nucleotide variations were detected in the putative keratin-associated protein 2-1 gene variants (Figure 4). Three of these (c.-103C>T, c.-70G>A, and c.-25G>A) were located in the 5′ untranslated region (5′-UTR), and six (c.42C>T, c.142T>C, c.192C>T, c.264G>C, c.303C>T, and c.348G>T) were in the coding region. Of the six coding region nucleotide variations, only one (c.142T>C) was non-synonymous and would result in a putative amino acid change (p.Ser48Pro).

From c.43 to c.347 (spanning the nucleotide variation c.142T>C, c.192C>T, c.264G>C, and c.303C>T), variants *A* and *C* had identical nucleotide sequences, whereas variants *B* and *D* were different. A chi sequence (5′-CCACCAGC-3′, reverse complementarity to 5′-GCTGGTGG-3′) occurred at positions c.332 to c.339. Three chi-like sequences from c.57 to c.64 (5′-CCTCCAGC-3′, reverse complementary to 5′-GCTGGAGG-3′), from c.275 to c.282 (5′-GCTGCTGG-3′), and from c.362 to c.369 (5′-CCACCTGC-3′, reverse complementary to 5′-GCAGGTGG-3′) were detected in the four sequence variants (Figure 4).

### 3.3. Amino Acid Sequence Analyses

The four ovine keratin-associated protein 2-1 gene sequences would all encode polypeptides of 132 amino acids in length (Figure 5). These polypeptides contained a high level of cysteine (23.48%), and moderate contents of proline (13.64%–4.39%), threonine (11.4%), arginine (10.60%), and serine (9.09%–9.85%). Other amino acid residues present in the polypeptides included valine, glutamine, glycine, alanine, phenylalanine, leucine, aspartic acid, glutamic acid, isoleucine, tyrosine, methionine, and tryptophan, and these accounted for 6.82, 5.30, 4.55, 2.27, 2.27, 2.27, 1.52, 1.52, 1.52, 1.52, 0.76, and 0.76 mole percent, respectively. Histidine, lysine, and asparagine were absent from the polypeptides. The theoretical isoelectric points (pI) of these four polypeptides were all 8.61.

### 3.4. Frequencies of the Keratin-Associated Protein 2-1 Gene Variants in Four Chinese Sheep Breeds

The frequencies of the ovine keratin-associated protein 2-1 gene variants in the 216 sheep are listed in Table 1. Variant *C* was the most common in all of the four breeds investigated, with a frequency ranging from 55.3% in the Chinese Merino sheep to 79.0% in the Tibetan sheep. In the Tibetan sheep and the Kazakh sheep, *B* was the second most common variant (with a frequency of 15.3% and 16.0%, respectively), and variants *A* and *D* were found to be the least common. However, in the Chinese Merino and Gansu Alpine Fine-wool sheep, *B* was the least common variant and the second most common variant was *A* (with a frequency of 19.7% and 22.7% in the Chinese Merino and the Gansu Alpine Fine-wool sheep, respectively).

### 3.5. Tissue Expression of Keratin-Associated Protein 2-1 Gene

The reverse transcription quantitative PCR analysis of the seven different tissues collected from Gansu Alpine Fine-wool sheep revealed that keratin-associated protein 2-1 gene was not only expressed in skin, but also found to be expressed in all of the other six tissues, including longissimus dorsi muscle, kidney, lung, spleen, heart, and liver tissue (Figure 6). To test whether this resulted from cross-contamination with the skin sample, we tested the expression of another high-sulfur keratin-associated protein gene, the keratin-associated protein 15-1 gene, using the same complementary deoxyribonucleic acid samples. Only the skin complementary deoxyribonucleic acid sample was found to be positive for keratin-associated protein 15-1 gene expression (Figure 7). To further verify the reverse transcription quantitative polymerase chain reaction result, we then tested whether the complementary deoxyribonucleic acid samples were contaminated with genomic deoxyribonucleic acid. As all of the keratin-associated protein genes are intron-less, we chose the intron-containing ADIPOQ gene for testing. None of the complementary deoxyribonucleic acid samples were polymerase chain reaction-positive for the ADIPOQ gene (Figure 7), confirming that deoxyribonucleic acid contamination was unlikely to be an issue.

The level of expression of keratin-associated protein 2-1 gene in the various tissues was investigated with reverse transcriptase-quantitative polymerase chain reaction analysis, revealing that ovine keratin-associated protein 2-1 gene had the highest level of expression in skin, and had a relatively high level of expression in heart and lung. It was expressed at lower levels in kidney tissue, longissimus dorsi muscle, liver, and spleen (*p* < 0.05) (Figure 8).

## 4. Discussion

This study identified a new ovine keratin-associated protein gene encoding a high-sulfur keratin-associated protein. The gene was clustered with several known keratin-associated protein genes on sheep chromosome 11, was located at a previously unannotated position, and had a predicted amino acid sequence most similar to the ovine keratin-associated protein 2-1 sequence, together suggesting that it represents the ovine homologue of the keratin-associated protein 2-1 gene. This is supported by it occurring at a physical location that matches with the location of the human keratin-associated protein 2-1 gene between the human keratin-associated protein 1 and keratin-associated protein 4 gene families [12].

Despite the predicted ovine keratin-associated protein 2-1 sequences having similarity to human keratin-associated protein 2-1, there were some differences in polypeptide sequence between the two species. First was the length of the proteins. The putative ovine keratin-associated protein 2-1 would be 132 amino acids in length, while the human protein is 128 amino acids in length, with the human protein having a four amino acid deletion at the carboxy-terminal end of the polypeptide. Between species differences in length have been observed for other keratin-associated proteins such as keratin-associated protein 15-1 [22] and keratin-associated protein 13-3 [23]. The second difference between the putative sheep keratin-associated protein 2-1 and the human protein was in the number of amino acid repeat sequences. It has been observed that keratin-associated protein 2-1 proteins possess cysteine-rich pentameric repeat structures (CCXPX) [2], but the frequency of occurrence of these repeat structures in the putative ovine keratin-associated protein 2-1 was different to that found in humans, with six repeats in sheep compared to five in the human sequence (Figure 5). The third difference between the putative sheep keratin-associated protein 2-1 and the human protein was in the content of cysteine. The ovine keratin-associated protein 2-1 protein putatively contains 23.48 mole percent of cysteine, which is lower than that present in the human sequence (27.34 mole percent) [24]. Finally, the ovine keratin-associated protein 2-1 protein would possess a carboxyl-terminal sequence different to that found in the human keratin-associated protein 2-1 protein. It has been suggested that differences in the sequence, length, and number of keratin-associated protein 2-1 genes between species may account for different hair/fiber phenotypes in different mammals [25], but the consequences of these differences between the putative ovine keratin-associated protein 2-1 and human keratin-associated protein 2-1 require further investigation.

The high pI value (8.61) predicted for ovine keratin-associated protein 2-1 suggests that the protein may exist in a basic form under normal physiological conditions. This protein may have a greater affinity for type I (acidic) keratins on a charge-only basis, with this being supported by the observation of Fujikawa et al. [12], who reported that human keratin-associated protein 2-1 binds to the type I hair keratin K34. A high pI value has been observed for other high-sulfur keratin-associated proteins, including keratin-associated protein 15-1 [22], 24-1 [10], and 13-3 [23].

It is noteworthy that most of the nucleotide variation found in the ovine keratin-associated protein 2-1 gene sequence were either located in non-coding regions or were synonymous, with only one (out of nine) being non-synonymous. This suggests that the ovine keratin-associated protein 2-1 may be under some structural constraint or subject to a selective pressure. A predominance of synonymous variation has also been reported for the ovine keratin-associated protein 1-2 gene [2] and the keratin-associated protein 1-3 gene [26], while variation in the ovine keratin-associated protein 1-4 gene [27] is predominantly non-synonymous. This suggests that the nucleotide sequence variation in the ovine keratin-associated protein 2-1 gene may have originated from similar mechanisms to those creating diversity in the keratin-associated protein 1-2 gene and the keratin-associated protein 1-3 gene, but different to the ovine keratin-associated protein 1-4 gene. This might be supported by the physical location of the genes on the chromosome where the keratin-associated protein 2-1 gene is physically closer to the keratin-associated protein 1-2 gene and the keratin-associated protein 1-3 gene than the keratin-associated protein 1-4 gene (Figure 1).

The putative keratin-associated protein 2-1 protein does not contain a high level of glycine, but has higher levels of serine and threonine. In the high-sulfur keratin-associated proteins keratin-associated protein 11-1 [28], keratin-associated protein 13-3 [23], and keratin-associated protein 24-1 [10], many of the serine and threonine residues are predicted to be phosphorylated. Despite phosphorylation having not been physically confirmed for the keratin-associated proteins, it has been described for the keratins, and it appears to affect protein assembly and organization [29]. The non-synonymous nucleotide variations (c.142T>C) would result in the gain or loss of serine, which may lead to functional effects if that serine was phosphorylated. This requires further investigation.

Although the five synonymous nucleotide variations and three nucleotide variations in the 5’-UTR identified in the study would not change amino acid sequences, the function of these nucleotide variations should not be ignored, as they may affect gene expression and protein folding [30,31]. In addition, these nucleotide sequence variations may be linked to other variations elsewhere in the ovine keratin-associated protein 2-1 gene, which may affect gene function and expression.

Although the association between variation in the ovine keratin-associated protein 2-1 gene and wool traits could not investigated in this study due to the lack of wool phenotypic data, we observed differences in variant frequencies between the fine-wool and coarse-wool sheep breeds. This suggests that the keratin-associated protein 2-1 gene may have been under selection pressures and/or plays a role in regulating wool fiber diameter. This effect appears to be different to that reported for the nearby keratin-associated protein 1-2 gene for which the association was detected for wool weight traits [8]. Overall, this suggests that different keratin-associated protein genes may have effects on different wool traits, despite the genes being clustered.

It is interesting to note that the ovine keratin-associated protein 2-1 gene not only has a high level of expression in skin, but also exhibits moderate levels of expression in heart and lung tissue, as well as low levels of expression in all the other tissues investigated. This is in contrast to many other keratin-associated protein genes investigated in sheep and goats, including the caprine keratin-associated protein genes 15-1 [20], 20-1 [30], and 20-2 [32], and the ovine keratin-associated protein genes 3-3 and 11-1 [33], for which the expression is exclusively found in wool follicles/skin. There is, however, one study suggesting that the caprine keratin-associated protein 11-1 gene is not only expressed in skin, but also in liver and heart [34]. In humans, while the keratin-associated protein 2-1 gene is reported to be highly expressed in the hair shaft cortex [12], expression of the gene is also described in many other tissues/organs, with high levels of expression reported for lung, esophagus, skin, kidney, and heart tissues in the online database [35]. This appears to suggest that the keratin-associated protein 2-1 may have other functions in addition to its potential role in wool fiber development, but further research is needed to confirm this. A separate gene expression study of sheep of differing age, given the sheep studied here were two-years old and thus not fully mature, as well as also including more tissues, would be a useful starting point.

## Figures and Tables

**Figure 1 genes-11-00604-f001:**
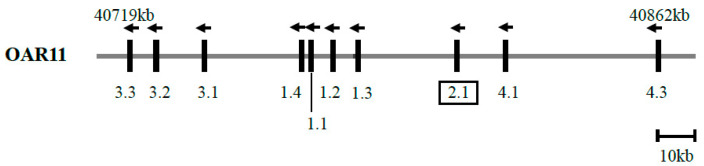
The location of the keratin-associated protein 2-1 gene on ovine chromosome 11. The newly identified keratin-associated protein 2-1 gene is boxed, and nine previously identified keratin-associated protein genes located nearby are also shown. A vertical bar indicates a keratin-associated protein gene, the arrow above the bar represents the direction of transcription, and the number below the bar represents the specific keratin-associated protein gene identity (e.g., 2.1 represents the keratin-associated protein 2-1 gene). The nucleotide coordinates refer to the Ovine Genome Assembly version 4.0, and are only approximate.

**Figure 2 genes-11-00604-f002:**
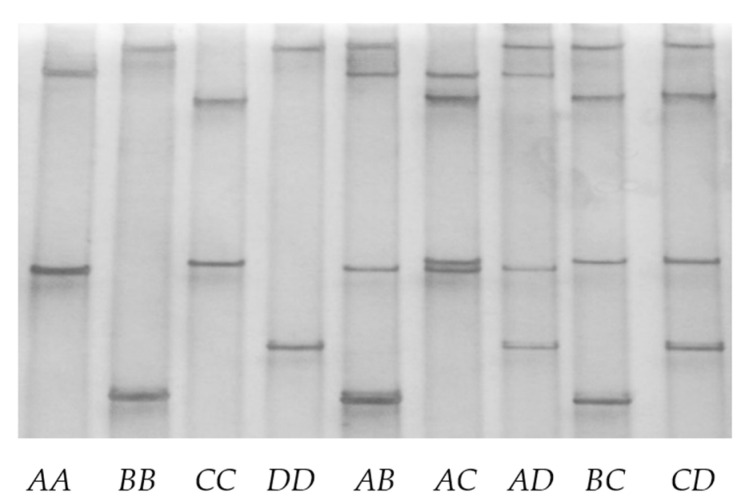
Variation in ovine keratin-associated protein 2-1 gene detected with polymerase chain reaction–single strand conformation polymorphism analysis. Four different banding patterns representing four variants (*A–D*) are shown in either homozygous (*AA, BB, CC* and *DD*) or heterozygous (*AB, AC, AD, BC* and *CD*) forms.

**Figure 3 genes-11-00604-f003:**
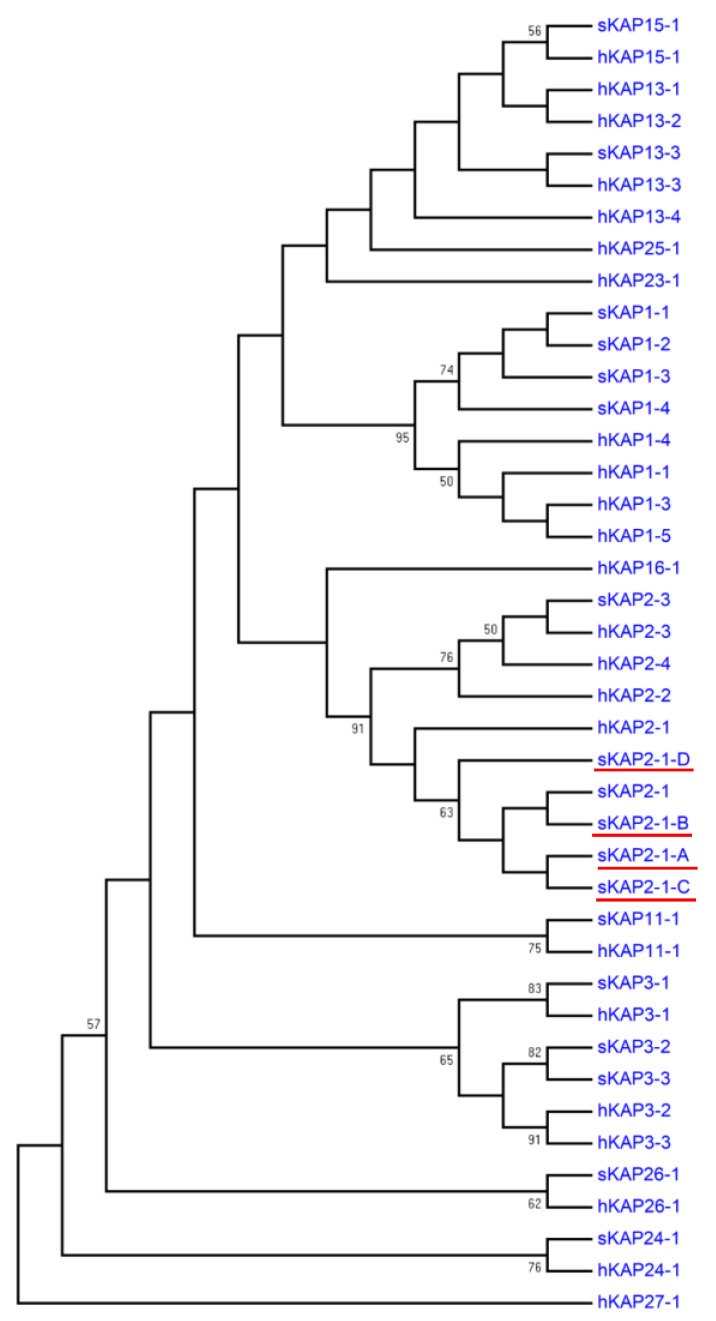
Phylogenetic tree of high-sulfur keratin-associated proteins identified in sheep and humans. The tree was constructed using amino acid sequences. The numbers at the forks represent the bootstrap confidence values and only those equal to or higher than 50% are shown. The sheep sequences are indicated with “s” and those from humans with “h”. The four newly identified sheep keratin-associated protein 2-1 sequences are indicated with red horizontal lines. The GenBank accession numbers for other high-sulfur keratin-associated proteins are X01610 (sKAP1-1 and sKAP1-4), NM_030967.2 (hKAP1-1), HQ897973 (sKAP1-2), X02925 (sKAP1-3), NM_030966.1 (hKAP1-3), NM_001257305.1 (hKAP1-4), NM_031957.1 (hKAP1-5), P02443 (sKAP2-1), NM_001123387.1 (hKAP2-1), NM_033032.2 (hKAP2-2), P02441 (sKAP2-3), NM_001165252.1 (hKAP2-3), NM_033184.3 (hKAP2-4), P02446 (sKAP3-1), NM_031958.1 (hKAP3-1), P02444 (sKAP3-2), NM_031959.2 (hKAP3-2), P02445 (sKAP3-3), NM_033185.2 (hKAP3-3), HQ595347 (sKAP11-1), NM_175858.2 (hKAP11-1), NM_181599.2 (hKAP13-1), NM_181621.3 (hKAP13-2), JN377429 (sKAP13-3), NM_181622.1 (hKAP13-3), NM_181600.1 (hKAP13-4), KX817979 (sKAP15-1), NM_181623.1 (hKAP15-1), NM_001146182.1 (hKAP16-1), NM_181624.1 (hKAP23-1), JX112014 (sKAP24-1), NM_001085455.2 (hKAP24-1), NM_001128598.1 (hKAP25-1), KX644903 (sKAP26-1), NM_203405.1 (hKAP26-1), and NM_001077711.1 (hKAP27-1).

**Figure 4 genes-11-00604-f004:**
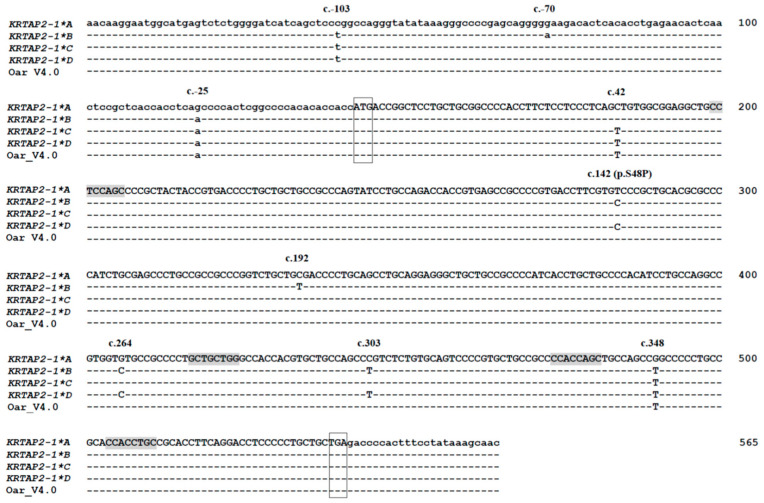
Alignment of the ovine keratin-associated protein 2-1 gene variant sequences (*A*–*D*), together with the Ovine Genome Assembly version 4.0. The abbreviation *KRTAP2-1* stands for keratin associated protein gene 2-1. Nucleotides outside the coding region are shown in lowercase, while those in the coding region are in uppercase. The predicted ATG start codon and TGA stop codon are marked with boxes. The positions of single nucleotide variations are shown above the sequences, and the non-synonymous substitution (p.S48P) is indicated. Dashes represent nucleotides identical to the top sequences. Chi and chi-like sequences are shaded. The nucleotide and amino acids are numbered according to the guidelines of the Human Genome Variation Society (HGVS) nomenclature, with, for example, c.-25 being the nucleotide position 25 base pairs upstream of the ATG start codon.

**Figure 5 genes-11-00604-f005:**
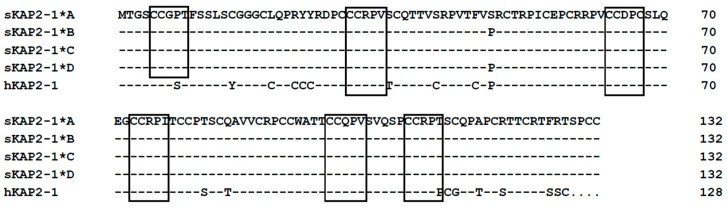
Alignment of the predicted amino acid sequences derived from keratin-associated protein 2-1 gene from sheep, and the human protein sequence. Dashes are used to indicate the same amino acids as the top sequence, while dots have been introduced to improve the alignment. The sheep and human sequences are indicated with the prefix “s” and “h”, respectively, and KAP2-1 represents the keratin-associated protein 2-1. The cysteine-containing pentameric repeat structures are boxed.

**Figure 6 genes-11-00604-f006:**
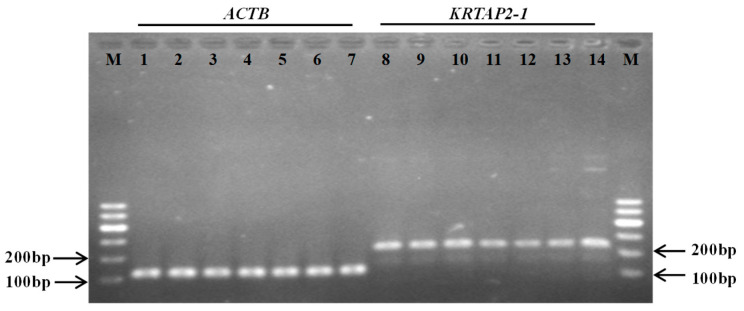
Expression of the keratin-associated protein 2-1 gene in different tissues derived from Gansu Alpine Fine-wool sheep. The keratin-associated protein 2-1 gene transcript was detected in all of the tissues investigated (lanes 8–14) using the beta-actin gene (*ACTB*) as an internal reference standard (lanes 1–7). M: 100 bp deoxyribonucleic acid ladder; lanes 1 and 8: skin; lanes 2 and 9: heart; lanes 3 and 10: liver; lanes 4 and 11: spleen; lanes 5 and 12: lung; lanes 6 and 13: kidney; lanes 7 and 14: longissimus dorsi muscle.

**Figure 7 genes-11-00604-f007:**
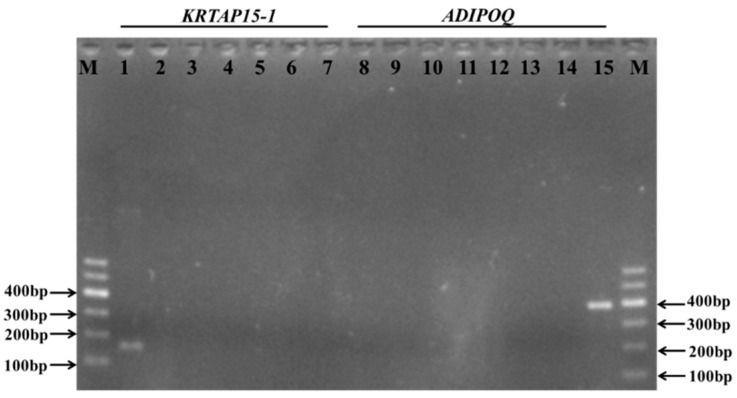
Polymerase chain reaction amplification of keratin-associated protein 15-1 gene and *ADIPOQ* using complementary deoxyribonucleic acid samples derived using reverse transcriptase from ribonucleic acid from seven tissues obtained from Gansu Alpine Fine-wool sheep. M: 100 bp deoxyribonucleic acid ladder; lanes 1 and 8: skin; lanes 2 and 9: heart; lanes 3 and 10: liver; lanes 4 and 11: spleen; lanes 5 and 12: lung; lanes 6 and 13: kidney; lanes 7 and 14: longissimus dorsi muscle; lane 15: positive control using a sheep genomic DNA sample.

**Figure 8 genes-11-00604-f008:**
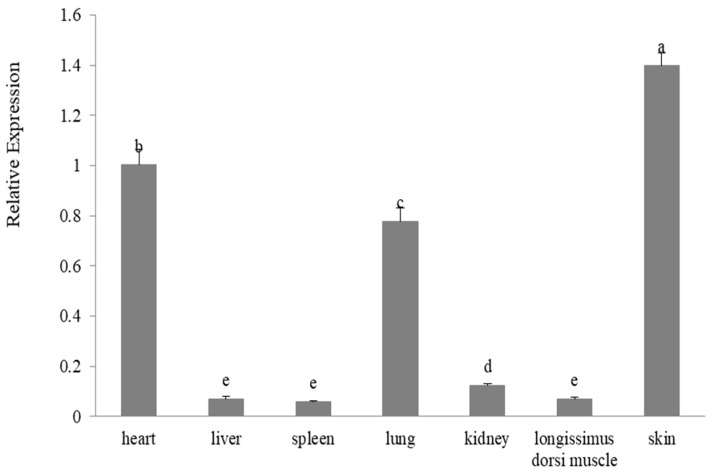
The relative expression levels of the ovine keratin-associated protein 2-1 gene in the seven tissues obtained from Gansu Alpine Fine-wool sheep. Different lowercase letters above the bars indicate that significant differences (*p* < 0.05) in expression of keratin-associated protein 2-1 gene exist between tissues, with, for example, skin having a significant difference in keratin-associated protein 2-1 gene expression relative to all the other tissues studied.

**Table 1 genes-11-00604-t001:** Frequencies of ovine keratin-associated protein 2-1 gene variants found in four Chinese sheep breeds.

Breed	*n*	Variant Frequencies (%)			
		*A*	*B*	*C*	*D*
Tibetan sheep	62	3.2	15.3	79.0	2.5
Kazakh sheep	50	7.0	16.0	74.0	3.0
Chinese Merino sheep	38	19.7	9.2	55.3	15.8
Gansu Alpine Fine-wool sheep	66	22.7	3.1	65.9	8.3

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
