# Peer review of "Identification of the Ovine Keratin-Associated Protein 2-1 Gene and Its Sequence Variation in Four Chinese Sheep Breeds"

_genes, 2020, doi:10.3390/genes11060604_

Round 1

Reviewer 1 Report

The work entitled “Identification of the ovine keratin-associated protein KAP2-1 gene (KRTAP2-1) and its sequence variation in four Chinese sheep breeds” by Wang et al. described about KAP2-1 gene and its sequence variation. It is interesting but the authors need to address the following questions. In materials methods (lines 77-79), the authors stated “A total of 216 sheep randomly selected from two fine-wool sheep breeds and two coarse-wool sheep breeds were investigated. These included Chinese Merino sheep (n = 38), Gansu Alpine Fine-wool sheep (n = 66), Tibetan sheep (n = 62) and Kazakh sheep (n = 50).” It is important to mention in methods section about the two sheep breeds belong to fine-wool and the other two belong to coarse-wool. For example, I understand Gansu Alpine belong to fine-wool sheep, but what about others? Are there any reasons to select different number of sheep for different groups? For example, Merino sheep only 38, but the others three are equal to or more than 50. The authors stated in methods section (lines 81-82) “Three separate two-year-old Gansu Alpine Fine-wool sheep were slaughtered to collect seven tissue samples from each; including skin, longissimus dorsi muscle, kidney, lung, spleen, heart and liver tissue.” Is it reasonable to study gene expression levels from two-year-old sheep? At this age, generally, growth-dependent gene expression occurs – it is important to discuss this issue in discussion part. In Figure 8 legend, I am confused, provide more details on “Different lowercase letters indicate significant differences at P < 0.05.” In Figure 4 legend, describe about c.-103, c.-70 …. C-.348 for the readers who are not working in the field.

Author Response

The work entitled “Identification of the ovine keratin-associated protein KAP2-1 gene (KRTAP2-1) and its sequence variation in four Chinese sheep breeds” by Wang et al. described about KAP2-1 gene and its sequence variation. It is interesting but the authors need to address the following questions.

In materials methods (lines 77-79), the authors stated “A total of 216 sheep randomly selected from two fine-wool sheep breeds and two coarse-wool sheep breeds were investigated. These included Chinese Merino sheep (n = 38), Gansu Alpine Fine-wool sheep (n = 66), Tibetan sheep (n = 62) and Kazakh sheep (n = 50).” It is important to mention in methods section about the two sheep breeds belong to fine-wool and the other two belong to coarse-wool. For example, I understand Gansu Alpine belong to fine-wool sheep, but what about others?

Detail to address this has now been added.

Are there any reasons to select different number of sheep for different groups? For example, Merino sheep only 38, but the others three are equal to or more than 50.

There was no reason. That simply reflects the number of sheep that were easily available for study.

The authors stated in methods section (lines 81-82) “Three separate two-year-old Gansu Alpine Fine-wool sheep were slaughtered to collect seven tissue samples from each; including skin, longissimus dorsi muscle, kidney, lung, spleen, heart and liver tissue.” Is it reasonable to study gene expression levels from two-year-old sheep? At this age, generally, growth-dependent gene expression occurs – it is important to discuss this issue in discussion part.

That is a fair point, and arguably a separate gene expression study at different ages is necessary. A comment has been added to the discussion to this effect.

In Figure 8 legend, I am confused, provide more details on “Different lowercase letters indicate significant differences at P < 0.05.”

Detail has been added.

In Figure 4 legend, describe about c.-103, c.-70 …. C-.348 for the readers who are not working in the field.

Detail has been added.

Reviewer 2 Report

The manuscript requires thorough revision with regard to the abbreviations used throughout the text. Abbreviations should be avoided in the abstract as much as in the keywords. Abbreviations that appear throughout the text should be abbreviated. The gene sequences in the text could perhaps be tabulated so that the flow is not disturbed when reading. The location and duration of the experiments are not mentioned in the text. Statistics are also missing from the manuscript. Places like CA should not be abbreviated. Internet addresses should not be given in the text, but in references. It is not clear what the newly discovered gene sequences mean for the sheep. The results should be presented with the meaning better and clearer. The manuscript suffers from a lack of representation.    

Author Response

The manuscript requires thorough revision with regard to the abbreviations used throughout the text. Abbreviations should be avoided in the abstract as much as in the keywords. Abbreviations that appear throughout the text should be abbreviated.

We have reduced the number of abbreviations as much as we think is possible, removing such things as SNP, but we probably need to maintain KAP for keratin-associated protein. This is a well-recognized abbreviation for wool proteins. We have removed HS, UHS and HGT when referring to high-sulfur, ultra-high sulfur and high glycine-tyrosine KAPs, PCR-SSCP, RT-qPCR, KRTAPs, MFD, PF, MSL, MFC, and EDTA. PCR is in common usage and was retained.

The gene sequences in the text could perhaps be tabulated so that the flow is not disturbed when reading.

This is layout usually addressed at the galley-proof stage and while we would be happy to put such things as primer sequences in a table, this will increase the number of tables and figures.

The location and duration of the experiments are not mentioned in the text.

The location of the experimental work has been added, but we are unsure of how to describe the duration of the experiment, be it overall (about six months), or in it’s component parts.

Statistics are also missing from the manuscript.

The only statistics used were in analyzing the results of the RT-qPCR, and were described asThe gene ACTB was utilized as an internal reference standard and the relative expression levels of KRTAP2-1 in the seven tissues were calculated using a 2-DDCt method.’

Places like CA should not be abbreviated.

We have added Quebec, California, Massachusetts, and The United States of America.

Internet addresses should not be given in the text, but in references.

We will need to get the journal to make the change for the BLAST algorithm (www.ncbi.nlm.nih.gov/) when mentioned as they generate the reference numbering system.

It is not clear what the newly discovered gene sequences mean for the sheep. The results should be presented with the meaning better and clearer. The manuscript suffers from a lack of representation.

We are not really sure what is meant by these comments. We discuss the structure and likely function of the new gene at length in the discussion.  What representation is needed?

Round 2

Reviewer 2 Report

The manuscript has been sufficiently revised. The problems are still the same as already described. You shorten terms like KAP, then the abbreviation KRTAP is used. There are unclear abbreviations in the heading and keywords. The manuscript needs careful revision. The other points such as statistics, place of study, time of study were not incorporated into the manuscript. An internet address is also given in the discussions and not in the references as recommended.

Round 3

Reviewer 2 Report

The revised manuscript is about "Identification of the ovin kreatin-associated protein 2-1 gene and its sequence variation in four Chinese sheep breeds".

The manuscript requires further revions:

  1. Abstract after 1. sentence: This animal study was carried out from ... to ... in ... (location / city / country).
  2. Keywords: double words: revome one: keratin-associated protein 2-1gene
  3. Line 48-50: change to:

    "Of these, keratin-associated protein 1, 3, 11, 13, 15, 16, 23, and 27 belong to the high-sulfur-keratin-associated protein group."

  4. Line 74: change to: "In this animal study"...
  5. Material and Methods: You have to renumber the headings. You start with the heading: 2.1 Ethic statement: for the first paragraph and then 2.2. Heading: Study design and Location: for example: " The experiments of this animal study were conducted from....to.... at the laboratory...(town/country). After that you continue with "2.3 Sheep investigated and tissue samples"
  6. Line 112-117: change to:

  7. Based on the putative ovine keratin-associated protein 2-1 gene sequence identified above, two polymerase chain reactionprimers

                     5’-AACAAGGAATGGCATGAGTC-3’ and

                     5’-GTTGCTTTATAGGAAAGTGGG-3’

    were designed to amplify a 565-bp deoxyribonucleic acid fragment that would include the entire coding region of the putative ovine keratin-associated protein 2-1 gene.These primers were synthesized by the Takara Biotechnology Company Limited (Dalian, People's Republic of China).

  8. write out DNA and TNF in Line 118
  9. Line 128: change to: 95° C
  10. Line 130: change to: 37.5 : 1
  11. Line 131: change to: 29° C
  12. Line 140: write out MEGA
  13. Line 145: exact date: which day
  14. Line 150: write out: gDNA
  15. Line 150-166: change to:

    Reverse transcription was carried out using the Prime Script™ RT Reagent kit with gDNA Eraser (Perfect Real Time: Takara). The synthesized complementary deoxyribonucleic acidwas amplified using a pair of PCR primers

                                  5’-CCAGTGTCCAGCCAGACCAC-3’ and

                                  5’-GGGGACTGCACAGAGACG-3’

    located within the keratin-associated protein 2-1 gene coding region. The amplification was carried out under the same conditions and thermal profile used for the amplification of genomic deoxyribonucleic acid, but the amplified template changed from genomic deoxyribonucleic acidto 0.8 μL of the complementarydeoxyribonucleic acid. The beta actin gene (ACTB) was used as an internal reference standard, and amplified using the primers

                                   5’-AGCCTTCCTTCCTGGGCATGGA-3’and

                                   5’-GGACAGCACCGTGTTGGCGTAGA-3’.

    The complementarydeoxyribonucleic acidsamples were tested for the presence of a transcript for ovine keratin-associated protein 15-1gene using the polymerase chain reactionprimers

                                    5’-ATCTTCCGCAGTCCCTG-3’ and

                                    5’-GATGACCGGCAACTCCT-3’, and the method described in Zhao et al. [20]. They were also tested for potential contamination with genomic deoxyribonucleic acidusing a pair of ADIPOQprimers

                                    5’-ACAGCGTGGATCTGGGTTC-3’ and

                                    5’-CACAATTCACTTTCGGCTGC-3’)

    which are located in intron 1 and intron 2 respectively, as described by An et al. [21]. The amplification products were electrophoresed through 1.0% agarose gels and visualizedusing ethidium bromide staining.

    16. Line 170: write out PCR

    17. Line 171: change to: 71° C

    18. Line 172: write out: ACTB

    19. After Line 174: Subheading with renumbering: write out the abbreviations RT-qPCR, ACTB, KRTAB2-1, ANOVA etc:

    Statistic

The only statistics used were in analyzing the results of the RT-qPCR, and were described asThe gene ACTB was utilized as an internal reference standard and the relative expression levels of KRTAP2-1 in the seven tissues were calculated using a 2-DDCt method.’

Error bars on the expression levelgraphs were calculated using a one-way ANOVA in IBM SPSSStatistics version 24.0(IBM China Company Limited, Beijing, People's Republic of China).

            20. Line 184-188: change to: ...the keratin-associated protein gene 3-3, 3-2, 3-1,1-4,1-1,1-2,1-3, 4-1 and 4-3.

            21: Line 202 und 220: write out DNA

            22: Figure 6: write out: DNA, ACTB

            23: Figure 7: remove (PCR), write out DNA

            24: Line 433: remove (CCXPX)

            25: Line 492-494 change to: ...keratin-associated protein gene 15-1 [20], 20-1 [30], 20-2 [32], 3-3 and the 11-1 [33];
